# Lactosylated Albumin Nanoparticles: Potential Drug Nanovehicles with Selective Targeting Toward an In Vitro Model of Hepatocellular Carcinoma

**DOI:** 10.3390/molecules24071382

**Published:** 2019-04-09

**Authors:** Nayelli Guadalupe Teran-Saavedra, Jose Andre-i Sarabia-Sainz, Erika Silva-Campa, Alexel J. Burgara-Estrella, Ana María Guzmán-Partida, Gabriela Ramos-Clamont Montfort, Martín Pedroza-Montero, Luz Vazquez-Moreno

**Affiliations:** 1Centro de Investigacion en Alimentacion y Desarrollo, A.C. Carretera Gustavo E. Aztiazaran 46, Hermosillo 83304, Sonora, Mexico; naye_krebs@hotmail.com (N.G.T.-S.); gupa@ciad.mx (A.M.G.-P.); gramos@ciad.mx (G.R.-C.M.); 2Departamento de Investigacion en Fisica. Universidad de Sonora, P.O. Box 5-088, Hermosillo, C.P. 83190, Mexico; andreisarabia@gmail.com (J.A.S.-S.); erika.silva@difus.uson.mx (E.S.-C.); alexel.burgara@gmail.com (A.J.B.-E.); mpedroza@cifus.uson.mx (M.P.-M.)

**Keywords:** nanoparticles, BSA-lactosylate, HepG2 cell line, asialoglycoprotein receptor

## Abstract

Hepatocellular carcinoma (HCC) ranks fifth in occurrence and second in mortality of all cancers. The development of effective therapies for HCC is urgently needed. Anticancer drugs targeted to the liver-specific asialoglycoprotein receptors (ASGPRs) are viewed as a promising potential treatment for HCC. ASGPRs facilitate the recognition and endocytosis of molecules, and possibly vehicles with galactose end groups, by the liver. In this study, bovine serum albumin (BSA) was conjugated with lactose using a thermal treatment. The formation of lactosylated BSA (BSA-Lac) was confirmed by a change of the chemical structure, increased molecular mass, and *Ricinus communis* lectin recognition. Subsequently, the low-crosslinking BSA-Lac nanoparticles (LC BSA-Lac NPs) and high-crosslinking BSA-Lac nanoparticles (HC BSA-Lac NPs) were synthesized. These nanoparticles presented spherical shapes with a size distribution of 560 ± 18.0 nm and 539 ± 9.0 nm, as well as an estimated surface charge of −26 ± 0.15 mV and −24 ± 0.45 mV, respectively. Both BSA-Lac NPs were selectively recognized by ASGPRs as shown by biorecognition, competition, and inhibition assays using an *in vitro* model of HCC. This justifies pursuing the strategy of using BSA-Lac NPs as potential drug nanovehicles with selective direction toward hepatocellular carcinoma.

## 1. Introduction

Hepatocellular carcinoma (HCC) is a public health concern because HCC is fifth in the occurrence rate for all cancers, and second only to lung cancers in mortality rate [1,2]. HCC incidence continues to increase with approximately 700,000 cases per year worldwide. The issue of greatest concern is the poor survival rate [3,4]. HCC is usually diagnosed in the advanced stages when prognosis for the disease recovery is meager, and survival is one to two months [5,6]. Until now, conventional treatment has been unsatisfactory for patients in this stage. Therefore, the development of effective therapies is urgently needed [2,7]. Recent reports suggest that targeting drugs to a specific tissue allows high concentrations of the drug within a tumor; this results in high efficacy and low toxicity in the organism [2,7]. In this sense, nanovehicles targeted to tumor sites could potentially be used as anticancer therapies with greater safety and efficacy [1,2]. 

In targeting drugs to a specific site, the selection of an appropriate surface receptor is crucial [1]. Furthermore, if receptor-mediated endocytosis follows receptor binding, this entry method could be used to direct drug-containing nanovehicles to any type of target cell [8]. The asialoglycoprotein receptor (ASGPR) is abundantly expressed on hepatocyte membranes (500,000 ASGPR/hepatocyte) and minimally expressed in extrahepatic tissues [4,6]. *In vivo*, the ASGPR recognizes and captures proteins and peptides that carry exposed galactose residues [4] and the receptor cargo enters the cell via receptor-mediated endocytosis.

In addition, enhanced permeability of the tumor vasculature allows nanovehicles to move in the tumor, whereas the suppressed lymphatic filtration allows them to be retained [9,10]. Hence, the nanoparticles retained in the tumor will interact with the ASGPR and initiate the process of endocytosis, achieving a targeted drug delivery.

The material of choice for the formation of nanostructures for human cell targeting is not a trivial issue, since it must comply with several safety requirements. Albumin is a globular protein that has been used as a therapeutic vehicle for anticancer drugs (such as doxorubicin) and that, due to effective permeability and retention (EPR), accumulates efficiently in tumors, improving the pharmacokinetic profile of anticancer agents [11,12,13]. Abraxane® is an example of this type of drug; it is an albumin conjugate linked to paclitaxel that has been approved by the Food and Drug Administration (FDA) for the treatment of metastatic breast cancer [14]. Thus, albumin is considered potentially useful for the synthesis of drug nanovehicles.

Previously, we reported the modification of bovine serum albumin (BSA) with lactose through thermal glycation. Lactose was coupled to the amino groups of lysine residues present in the BSA, while galactose remained unchanged. Using this method, we obtained lactosylated bovine serum albumin (BSA-Lac) as the product [15,16]. Lactose has several advantages; it lacks immunogenicity, it is highly stable, and is easy to use for modification. Lactose is considered an excellent ligand for several artificial systems [1,6,17] and a promising candidate for the nanovehicle targeting of drugs to the ASGPR [8,17]. The aim of the present work was to synthesize BSA-Lac NPs and test their specific biorecognition by ASGPRs present on HepG2 cells, which is a line derived from an HCC.

## 2. Results and Discussion

### 2.1. Characterization of the BSA-Lac

#### 2.1.1. Biorecognition Assays of BSA-Lac with RCA

The thermal glycation of BSA with the disaccharide lactose (galactose-β (1-4) glucose) results in the covalent binding of the reducing sugar (glucose) with the amino groups of BSA. Albumin glycated with lactose (BSA-glucose-β (4-1) galactose) provides galactose residues that are available for biological recognition by specific receptors such as those involved in bacterial adhesins or plant lectins [15,18]. We evaluated the BSA-Lac for exposed galactose sites that could be recognized by receptors using a semiquantitative enzyme-linked lectin recognition assays (ELLA). *Ricinus communis agglutinin* (RCA) lectin was used, because it is known to react specifically with galactose. Figure 1 shows a strong interaction of RCA with the BSA-Lac when compared with BSA thermally treated in the absence of lactose (tBSA), and demonstrates the specific recognition of the RCA for BSA-Lac. These results confirm that, as expected, the thermal glycation resulted in galactose sites that retained sufficient structure and availability for biorecognition.

#### 2.1.2. Electrophoresis SDS-PAGE

SDS-PAGE electrophoresis was used to analyze the molecular mass of BSA-Lac, using tBSA as control. The molecular weight of tBSA (Figure 2, lane A) was 66.2 kDa, while that of BSA-Lac (Figure 2, lane B) was estimated as 71.2 kDa. The estimation of the relative molecular mass shows that the glycation of BSA induced a molecular weight difference of 5 kDa with respect to non-glycated BSA. The BSA has 60 lysines and 26 arginines with amino groups available to glycation reaction; the increase in mass in BSA-Lac (5 kDa) indicates an addition of about 14 molecules of lactose (324.3 Da molecular weight) per protein molecule. Studies in our laboratory have shown that albumin can be glycated with up to 41 galactose residues [15].

To further confirm that BSA was lactosylated, lectin blotting was conducted using RCA lectin. While RCA did not interact with tBSA, it presented a strong signal indicating the reaction with BSA-Lac, as shown in Figure 2, lanes C and D, respectively.

#### 2.1.3. FT-IR

Fourier transform infrared spectroscopy (FT-IR) analysis is an effective method to obtain information on the composition and chemical structure of molecules [19]. The structural changes of albumin due to the glycation process as well as the presence of lactose in the glycoconjugate were followed using FT-IR analysis. 

Figure 3 shows the infrared spectrum of BSA-Lac (red band) and compares it with that of the controls tBSA (purple band) and native BSA (green band). Differences were found in the range of 900 to 1199 cm^−1^ of wavelength, corresponding to the carbohydrate fingerprint (attributed to the presence of the CO bond) [20]. The BSA-Lac band shown has a higher intensity of vibration, as indicated by a higher and wider band [21]. Both controls, tBSA and the native BSA, presented similar spectra. The FT-IR results indicate that the heat treatment did not modify the chemical composition of the molecule, and that BSA was successfully conjugated with lactose to form BSA-Lac. 

Taken together, results from biorecognition assays of BSA-Lac with RCA, electrophoresis SDS-PAGE, and FT-IR indicated that BSA-Lac was suitable for the synthesis of nanoparticles with the possibility of the selective targeting of hepatic ASGPR.

### 2.2. Characterization of Nanoparticles

We prepared the nanoparticles using BSA-Lac and ethyl alcohol as the desolvating agent. Two concentrations of crosslinker (glutaraldehyde), low (LC) and high (HC), were used to test for their impact on biorecognition. Thus, we obtained four different nanoparticles: LC BSA-Lac NPs, HC BSA-Lac NPs, and as controls, LC tBSA NPs and HC tBSA NPs. These NPs were used to evaluate the size, charge, and morphology of nanoparticles, as well as their biorecognition by HepG2 cells. 

#### 2.2.1. Particle Size and Zeta Potential

The average sizes of NPs were obtained by dynamic light scattering (DLS) (Table 1). The size distribution was higher for the LC BSA-Lac NPs and HC BSA-Lac NPs with average sizes of 560 ± 18.0 nm and 539 ± 9.0 nm, respectively, while LC tBSA NPs and HC tBSA NPs had an average size of 241 ± 2.5 nm and 246 ± 4.4 nm, respectively. It has been reported that the glycation of BSA induces an increase in the isoelectric point (Ip). For example, Rubio-Ruiz et al. (2008) [22] reported that BSA glycated with glucose (48 glucoses per BSA molecule) had a change of Ip from 4.2 to 6.3. Ip is an important parameter for the control of nanoparticle size by the desolvation method. When the nanoparticles are prepared at a pH close to the isoelectric point, a higher coagulation of the protein occurs, and as a result, larger nanoparticles are obtained [23]. In this work, the synthesis of nanoparticles was done at pH 7, and the larger size of BSA-Lac NPs, relative to that of tBSA NPs, can be attributed to the isoelectric point of glycated albumin being close to the pH of the preparation. The sizes obtained from the BSA-Lac NPs with LC and HC were similar, and are suitable for the purpose of this work, because NPs greater than 100 nm can be endocyted by liver cells [24].

The zeta potential of BSA-Lac and tBSA NPs was measured to determine their surface charge (Table 1). This parameter is important for the stability of suspended particles [6], and the electrostatic interaction with the biological environment. The zeta potentials of the LC BSA-Lac NPs, HC BSA-Lac NPs, LC tBSA NPs, and HC tBSA NPs were determined as −26 ± 0.15 mV, −24 ± 0.45 mV, −30 ± 0.9 mV, and −27 ± 0.2 mV, respectively. These values indicate the good stability of the NPs in suspension and the absence of the formation of aggregates in a colloidal solution [25]. Reports by others indicate that negative charges on nanoparticles reduce interaction with plasma proteins, and thereby, decrease the probability of activating an immune response by monocytes in blood [26,27].

#### 2.2.2. Nanoparticle Morphology

We analyzed the morphology of HC and LC BSA-Lac and tBSA NPs using atomic force microscopy (AFM) (Figure 4A) and SEM (Figure 4B). All of the types of NPs presented a spherical shape with a smooth surface. The polydispersity of the NPs was also observed. The size diameter of the BSA-Lac NPs was approximately 500 nm, while that of tBSA was 200 nm. These results are within the polydispersity and dispersion range of sizes obtained by dynamic light scattering (DLS).

### 2.3. Evaluation of Specific Cellular Recognition

The cellular interaction of FITC-labeled nanoparticles was studied in two cell lines, human liver cancer (HepG2), which abundantly present ASGPR [28,29] and human cervical carcinoma (HeLa), which was used as a control, due to the absence of ASGPR [30,31]. The assays were performed by fluorescence confocal microscopy (Figure 5A) and the fluorescence intensity of cells exposed to NPs were quantitatively evaluated (Figure 5B). HepG2 cells incubated with either LC or HC BSA-Lac NPs showed higher fluorescence intensity, indicating that the galactose present in these NPs was recognized by the ASGPR. HepG2 cells incubated with tBSA NPs showed a fluorescence intensity around 3%, as compared with lactosylated NPs (Figure 5B). Although it has been reported that albumin is recognized by gp60, a glycoprotein expressed in the membrane of endothelial cells of tumor tissues [6,32], the reduced interaction of HepG2 cells receptors and tBSA NPs can be considered nonspecific.

To confirm the specific recognition of the BSA-Lac NPs with the ASGPR, competition and inhibition assays were done. In the competition assays, a moderate lactosylated uptake of NPs was observed (Figure 5A). The reduced fluorescence intensity (50%, Figure 5B) is because both the BSA-Lac NPs and free lactose are recognized by the HepG2 ASGPRs. Free lactose present in the media culture competed with the BSA-Lac NPs for the available receptors; thereby reducing BSA-Lac NP receptor binding and resulting in reduced fluorescence. In the inhibition assays, preincubation of the HepG2 cells with free lactose show a decreased uptake of BSA-Lac NPs (Figure 5A), reducing approximately 98% of the fluorescence intensity (Figure 5B). This indicate that previous exposure of the cells to lactose blocked BSA-Lac NP recognition and supported the notion that ASGPRs specifically recognizes the galactose present in BS-Lac NPs [8]. In contrast, as expected, no fluorescence was observed for HeLa cells, because the ASGPR is not expressed by this cell line.

## 3. Materials and Methods

### 3.1. Material

All of the reagents used were analytical-grade and, unless specified, all of the other reagents and chemicals were purchased from Sigma-Aldrich. Bovine serum albumin (BSA; 66.5 kDa and ∼96%), D-lactose monohydrate (Lac), glutaraldehyde (25%), Dulbecco’s modified Eagle’s medium (DMEM), fetal bovine serum (FBS), penicillin/streptomycin (P/S), fluorescein isothiocyanate (FITC), and [3-5-dimethylthiazol-2-yl]-2, 5-diphenyltetrazolium bromide (MTT) were purchased from Sigma-Aldrich (St. Louis, MO, USA). *Ricinus communis* agglutinin I (RCA I) was obtained from the Vector Lab (Burlingame, CA, USA). HepG2 (human liver cancer) and HeLa (human cervical carcinoma) cells were obtained from ATCC (American Type Culture Collection, Manassas, VA, USA). 

### 3.2. Lactosylation of Albumin

The modification of albumin with lactose was carried out according to Sarabia-Sainz et al. (2011) [22]. BSA-Lac samples were frozen at −40 °C and freeze-dried (Virtis Benchop 6.6, NY, USA), and subsequently incubated at 100 °C for 30 min. Finally, unreacted lactose was removed by dialysis. Thermally treated albumin without lactosylation was used as a negative control. Samples were freeze-dried and stored at −20 °C until later use.

### 3.3. Characterization of BSA-Lac

#### 3.3.1. Enzyme-linked Lectin Recognition Assays (ELLA)

The biorecognition of galactose was evidenced by the interaction of BSA-Lac with biotin-labeled RCA I. Briefly, 5 μg/100 μL BSA-Lac and negative controls (BSA) and BSA thermally treated in the absence of lactose (tBSA) were immobilized in a 96-well plate. Following adsorption, wells were blocked for 3 h with 20 mM of phosphate-buffered saline (PBS) containing, 0.05% Tween 20, pH 7.5 (PBS-T), and 1.5% BSA, to prevent nonspecific interactions. Next, 100 μL of biotinylated RCA was added at a concentration of 2.5 μg/mL, and incubated at room temperature for 1.5 h. After washed with PBS-T, samples were incubated with 100 μL of streptavidin-peroxidase (1:2000) in PBS for 40 min. The color of the reaction was developed using SIGMA FAST OPD, following the manufacturer’s instructions. Absorbance at 450 nm was read in an ELISA reader (Anthos Zenyth 340st, Alcobendas, Spain) at 10 min [15].

#### 3.3.2. SDS-PAGE Electrophoresis

BSA-Lac was analyzed by electrophoresis under denaturing and reducing conditions using 8% polyacrylamide gels (SDS-PAGE) according to Laemmli (1970) [33]. Controls included untreated BSA and tBSA. Each sample containing 5 µg of protein was loaded onto the gel and after running stained with 1% Coomassie blue. The relative molecular mass of the sample was estimated by comparison with molecular weight standards and documented using a Molecular Imager® Gel Doc™ XR+ System with Image Lab™ Software (Image Lab 3.0, BioRad, Hercules, CA, USA).

#### 3.3.3. Lectin-blotting Assay

Lectin-blotting assay was performed as described by Lagarda-Díaz et al. (2009) [34]. Proteins (BSA, tBSA, and BSA-Lac) previously separated by SDS-PAGE were transferred onto a nitrocellulose membrane at a rate of 0.8 mA/cm2 for 45 min, using a semi-dry blotter (LABCONCO, Kansas City, MO). Membranes were blocked for 1.5 h with PBS containing 2% BSA. BSA-Lac incubated with biotinylated RCA I (5 μg/mL for 3 h), followed by an incubation with streptavidin peroxidase (1: 2000) for 1.5 h. The color reaction was developed at room temperature by the addition of peroxidase substrate, 0.075% 3,3′-diaminobenzidine-4HCl (DAB).

#### 3.3.4. Fourier-transform Infrared Spectroscopy

The presence of functional groups in BSA-Lac and their absence in negative controls (BSA and tBSA) was evaluated by Fourier transform infrared (FT-IR) spectroscopy [17]. Infrared spectra were recorded using Agilent Cary 630 FTIR Spectrometer (Agilent, Cary 630 FTIR Spectrometer, Santa Clara, CA, USA) at a resolution of 4 cm−1 in the range of 650 to 4000 cm^−1^. 

### 3.4. Synthesis of BSA-Lac Nanoparticles (BSA-Lac NPs)

BSA-Lac NPs were prepared using a desolvation system to form nanoparticles as described by Gallegos-Tabanico et al. (2017) [14] with minor modifications. Briefly, 10 mg of BSA-Lac was dissolved in 1 mL of deionized water. Ethanol (3 mL) was added slowly dropwise into the BSA-Lac solution under constant stirring. Then, 5 or 10 μL of glutaraldehyde (8%) was added to each sample to induce the crosslinking of the BSA-Lac molecules and give stability to the formed nanoparticles. The mixture was stirred for 5 h and the nanoparticles were then washed three times with deionized water and recovered by centrifugation (1644× g for 10 min). Thus, low-crosslinking (5 μL of glutaraldehyde) BSA-Lac nanoparticles (LC BSA-Lac NPs), high-crosslinking (10 μL of glutaraldehyde) BSA-Lac nanoparticles (HC BSA-Lac NPs) were obtained. Also, for synthesis control, LC tBSA NPs, and HC tBSA NPs were prepared.

### 3.5. Characterization of BSA-Lac NPs

#### 3.5.1. Particle Size and Zeta Potential

The particle size (mean particle diameters and size distributions) and zeta potential of NPs were measured at 25 °C using dynamic light scattering (DLS) at a scattering angle of 90° with Zetasizer Nano ZS90 (Malvern Instruments Ltd, Malvern UK) with a doppler anemometry laser. Samples were diluted in PBS (1mg/mL) at pH 7.2. All of the measurements were done in triplicate. 

#### 3.5.2. Nanoparticle Morphology

The morphology of NPs was characterized by scanning electron microscopy (SEM, JEOL JSM-7800F, Akishima, Tokyo, Japan) and atomic force microscopy (AFM, XE-Bio system, Park Systems Corp, Su-won, Korea). SEM images were obtained with a magnification of 30,000× using an acceleration voltage of 5.0 kV. AFM images were reconstructed in the non-contact mode using Nanosensors: NCHR cantilevers (force constant 10 to 130 N/m). The analysis was performed using 5 × 5 μm scanning images. The three-dimensional (3D) images were analyzed with the software Gwyddion version 2.49 available online at http://gwyddion.net/.

### 3.6. Cellular Uptake Evaluation and Specificity of Recognition

Prior to evaluate the cellular interaction, NPs were labeled with FITC according to Huang et al. (2017) [4] with modification. Briefly, 0.5 mL of FITC (0.4 mg/mL DMSO) was mixed with 2 mL of NPs suspensión (5 mg). After 3 h of reaction in the dark at room temperature, the FITC-labeled NPs were washed using water and several centrifugation cycles (2422 ×g, 10 min at 25 °C). The conjugation of FITC with NPs was evaluated by the measurement of excitation and emission (λexc 480 and λemi 520 nm, respectively) spectra using a Fluorolog (Horiba JobinYvon, Palaiseau France) with the software FluorEssence 3.5.1.99.

NPs labeled with FITC were evaluated using Hep G2 (ASGPR+) cells and HeLa cells (ASGPR–) as control. Cells were seeded in 48-well plates at a density of 10,000 cells/200 µL per well using DMEM containing 10% FBS. After incubation (5% CO_2_ at 37 °C for 24 h), cells were washed three times with physiological saline solution (200 μL) and subsequently incubated with PBS containing BSA-Lac nanoparticles (LC BSA-Lac NPs, HC BSA-Lac NPs, LC tBSA NPs, or HC tBSA NPs) at a 10 μg/200 µL at 37 °C for 30 min. Cells were rinsed three times with physiological saline solution (200 μL), and then observed by confocal microscopy (the images dimensions were 1024 × 1024 pixels). For competition assays, cells were incubated simultaneously with NPs (10 μg/100 µL) plus or minus free lactose (10 μg/100 µL); for uptake inhibition assays, cells were preincubated with lactose (20μg/200µL) before the NPs (10 μg/200 µL) were added. The fluorescence intensity was quantified taking the cell body regions in the visual field that were selected as the regions of interest (ROI). The images and mean fluorescence intensity were obtained by confocal microscopy (Nikon TiEclipse C2+, Japan) with 488-nm lasers at 20x magnification.

### 3.7. Statistical Analysis

The results are presented as mean ± standard deviation. For ELLA, BSA, and BSA-Lac assay, the Student’s t-test was used. Statistical analysis of the fluorescence intensity of the cellular interaction was performed using a one-way ANOVA followed by the application of Tukey’s test. *p* ≤ 0.05 were considered statistically significant.

## 4. Conclusions

In summary, we obtained BSA-Lac NPs that were selectively bound to the ASGPR as demonstrated by our studies using an *in vitro* cell model of HCC. As noted above, ASGPRs are expressed almost exclusively by hepatocytes, and thus, this receptor can be used to facilitate the recognition and endocytosis of molecules or vehicles with exposed galactose groups specifically into the liver. Our results justify continued exploration into the strategy of using lactosylated albumin NPs as potential drug nanovehicles with selective direction toward the treatment of hepatocellular carcinoma. 

## Figures and Tables

**Figure 1 molecules-24-01382-f001:**
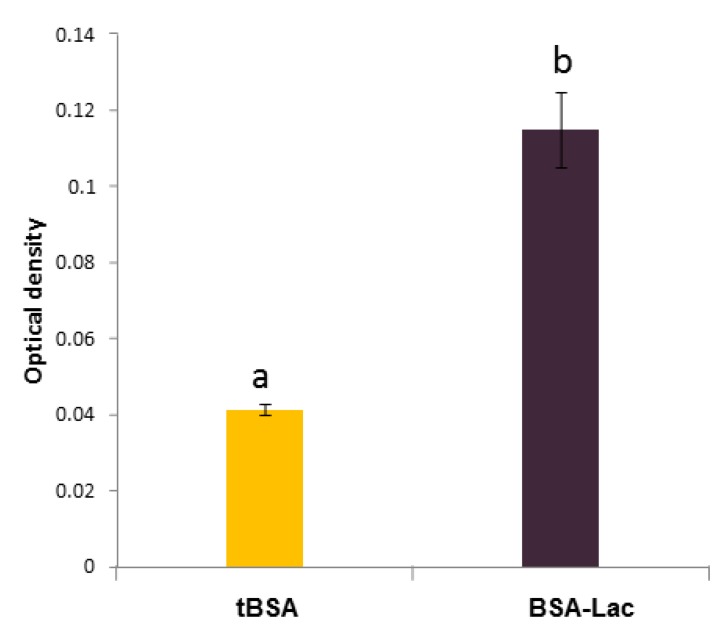
Enzyme-linked lectin recognition assays of lactosylated bovine serum albumin (BSA-Lac) and BSA thermally treated in the absence of lactose (tBSA). Optical density values were compared using the Student’s t-test. *p* ≤ 0.05. Different letters (a and b) indicate the statistical significance (*p* ≤ 0.05).

**Figure 2 molecules-24-01382-f002:**
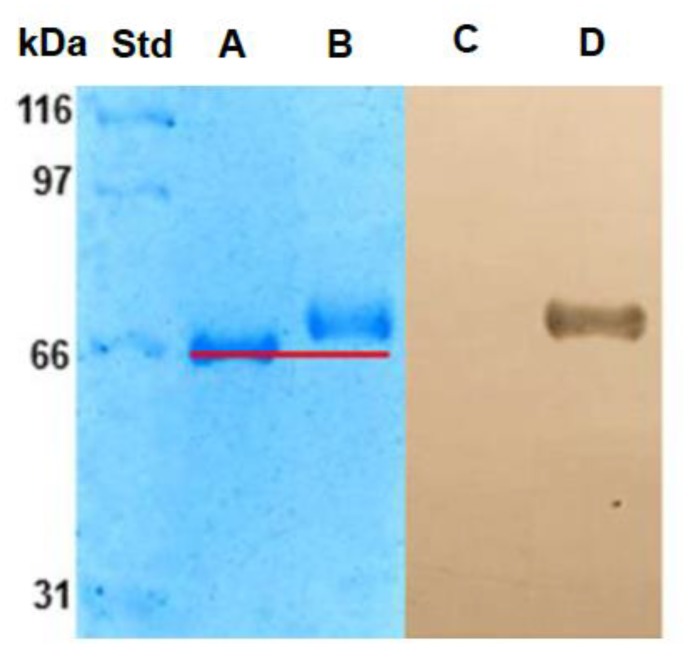
SDS-PAGE profiles and lectin blotting assay of BSA. Lane A, tBSA; Lane B, BSA-Lac; Lane C, tBSA and Lane D, BSA-Lac.

**Figure 3 molecules-24-01382-f003:**
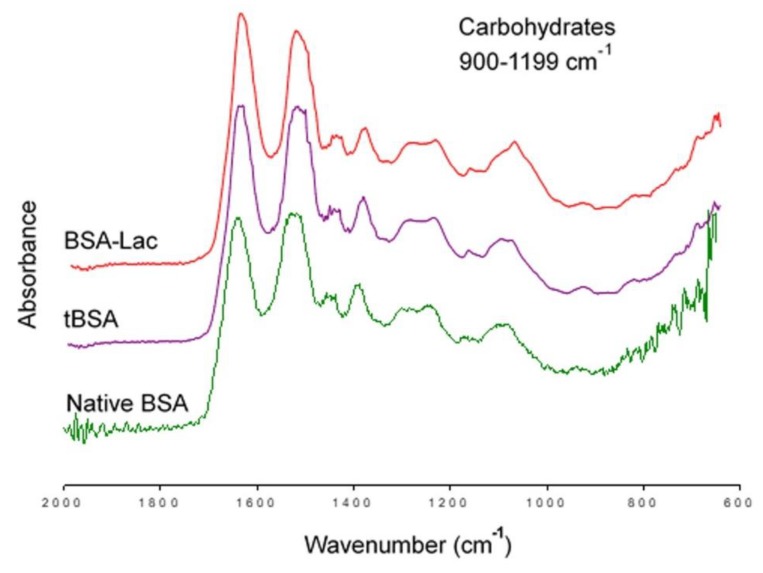
Fourier transform infrared spectroscopy (FT-IR) of BSA-Lac (red), tBSA (purple), and native BSA (green).

**Figure 4 molecules-24-01382-f004:**
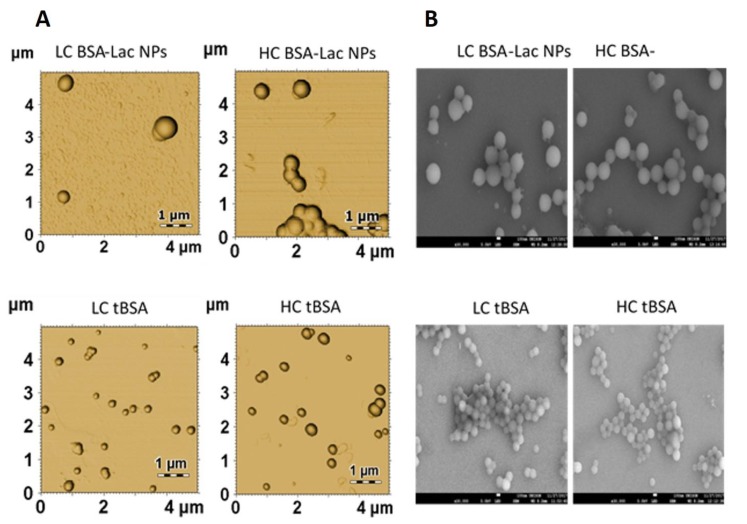
Analysis of nanoparticle morphology by (**A**) atomic force microscopy (AFM), and (**B**) scanning electron microscopy.

**Figure 5 molecules-24-01382-f005:**
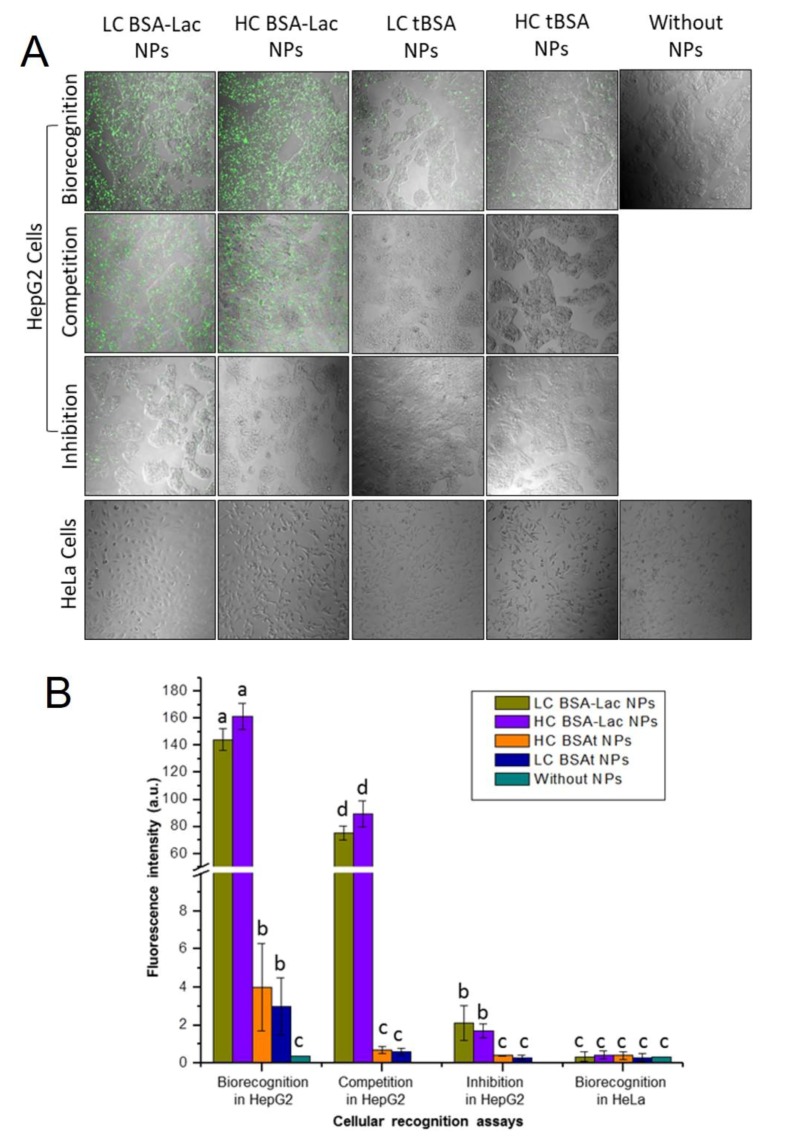
Evaluation of cellular interaction of NPs in human liver cancer (HepG2) and human cervical carcinoma (HeLa) cells. (**A**) Confocal fluorescence images of HepG2 and HeLa cells with LC or HC BSA-Lac NPs. NPs were labeled with fluorescein isothiocyanate (FITC). (**B**) Quantitative analysis of fluorescence intensity of HepG2 and HeLa cells exposed to NPs samples. Experiments were performed in triplicate and analysis by ANOVA followed by Tukey’s test; *p* ≤ 0.05. Different letter (a, b, c and d) shows statistical significance.

**Table 1 molecules-24-01382-t001:** Size and zeta potential of low-crosslinking (LC) BSA-Lac, high-crosslinking (HC) BSA-Lac, LC tBSA, and HC tBSA nanoparticles (NPs).

	NPs			
	LC BSA-Lac	HC BSA-Lac	LC tBSA	HC tBSA
Size (nm)	560 ± 18.0	539 ± 9.0	241 ± 2.5	246 ± 4.4
Zeta potential (mv)	−26 ± 0.1	−24 ± 0.4	−30 ± 0.9	−27 ± 0.2

Values are average and standard deviation (±) for triplicate.

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
