# Peer review of "Lactosylated Albumin Nanoparticles: Potential Drug Nanovehicles with Selective Targeting Toward an In Vitro Model of Hepatocellular Carcinoma"

_molecules, 2019, doi:10.3390/molecules24071382_

Reviewer 1 Report

The authors synthesized novel nanoparticles (NPs) targeting asialoglycoprotein receptor (ASGPR) on hepatocytes by lactosylation of bovine serum albumin (BSA-Lac). The synthesis of tumor-specific drug vehicles for the development of new targeted therapeutics is worth of investigation. However, the following points should be addressed.

< Major Point >

1. ASGPR is a C-type lectin, primary expressed on the sinusoidal surface of the hepatocyte. ASGPR is also abundantly expressed in normal hepatocytes, so it is difficult to say that it is a hepatocellular carcinoma (HCC) cell-specific molecule and it is inappropriate as a therapeutic target of HCC.

2. Experiments to confirm the expression of ASGPR in HepG2 cells and HeLa cells used in the experiments should be additionally provided.

3. The statement "Both BSA-Lac NPs 24 were selectively recognized by ASGPRs as demonstrated by competition assays using an in vitro model of HCC." (lines 24-26, Abstract) is key result of the present study. The experimental results supporting the statement should be described in detail.

4. The interpretation "Taken together, our studies indicated 99 that BSA-Lac was suitable for the synthesis of nanoparticles with the possibility of selective targeting 100 of hepatic ASGPR." (lines 99-101, the Results section) cannot be driven from the results of FT-IR analysis.

< Minor Points >

1. Statistical analyses for the experimental results is necessary. In addition, the statistical methodology needs to be described in the Methods section.

2. The abbreviation should be spelled out, in its first use in the text with the abbreviated form in parentheses

3. Major and minor grammatical errors are found throughout the text. English editing is absolutely required.

Author Response

Response to Reviewer 1 Comments

Comments and Suggestions for Authors

The authors synthesized novel nanoparticles (NPs) targeting asialoglycoprotein receptor (ASGPR) on hepatocytes by lactosylation of bovine serum albumin (BSA-Lac). The synthesis of tumor-specific drug vehicles for the development of new targeted therapeutics is worth of investigation. However, the following points should be addressed.

< Major Point >

1. ASGPR is a C-type lectin, primary expressed on the sinusoidal surface of the hepatocyte. ASGPR is also abundantly expressed in normal hepatocytes, so it is difficult to say that it is a hepatocellular carcinoma (HCC) cell-specific molecule and it is inappropriate as a therapeutic target of HCC.

Your statement is correct, however, is important to note that according to several reports in a tissue with tumor cells, there is a phenomenon of enhanced permeability and retention (EPR) which tends to accumulate certain molecules or particles in the tumor tissue. In general, this process is explained due to the rapid growth of tumor cells and the accelerated stimulation of blood vessels. The enhanced permeability of the tumor vasculature allows nanoparticles to enter in the tumor interstitial space. This phenomenon EPR has been the basis of nanotechnology platforms to deliver drugs to tumors. Hence, although ASGPR is abundantly expressed in normal hepatocytes, EPR would allow nanoparticles to interact with hepatoma receptors, generating greater impact,

In the manuscript, the following information was added to the introduction:

In addition, enhanced permeability of the tumor vasculature allows nano-vehicles to move in the tumor, whereas the suppressed lymphatic filtration allows them to be retained [9,10]. Hence, the nanoparticles retained in the tumor will interact with the ASGPR and initiate the process of endocytosis, achieving a targeted drug delivery.

Reports about EPR:

Maeda, H., Wu, J., Sawa, T., Matsumura, Y., & Hori, K. (2000). Tumor vascular permeability and the EPR effect in macromolecular therapeutics: a review. Journal of controlled release, 65(1-2), 271-284.

Prabhakar, U., Maeda, H., Jain, R. K., Sevick-Muraca, E. M., Zamboni, W., Farokhzad, O. C., ... & Blakey D. C. (2013). Challenges and key considerations of the enhanced permeability and retention effect for nanomedicine drug delivery in oncology. Cancer Research, 2412-2417.

Nel, A., Ruoslahti, E., & Meng, H. (2017). New insights into “permeability” as in the enhanced permeability and retention effect of cancer nanotherapeutics. ACS NANO, 9567-9569.

2. Experiments to confirm the expression of ASGPR in HepG2 cells and HeLa cells used in the experiments should be additionally provided.

The expression of ASGPR by cells has been reported only by genetic recombination processes (Schwartz et al., 1981; Zhang et al., 2016). The cells from cervical tissue naturally lacks ASGPR proteins. By the other hands several articles where demonstrated the presence and quantification of ASGPR in the HepG2 cell line. In this sense, we believe that the determination of ASGPR in the cells used in this study is not essential. In addition, the HeLa cell line where it was used as a negative control (absence of the ASGPR) (Spiess and Lodish, 1985; Hu et al., 2016).

In the manuscript, the following information was added to the evaluation of specific cellular recognition (results) section:

The cellular interaction of FITC-labeled nanoparticles was studied in two cell lines, HepG2 which has been reported in several publications the presence and abundance of ASGPR [28,29] and HeLa, which was used as a control, due to the absence of ASGPR [30,31]. The assays were performed by fluorescence and confocal microscopy (Figure 5).  

Reports that talk about the expression of ASGPR in HepG2 cells:

Schwartz, A. L., Fridovich, S. E., Knowles, B. B., & Lodish, H. F. (1981). Characterization of the asialoglycoprotein receptor in a continuous hepatoma line. Journal of Biological Chemistry, 256(17), 8878-8881.

Li, Y., Huang, G., Diakur, J., & Wiebe, L. I. (2008). Targeted delivery of macromolecular drugs: asialoglycoprotein receptor (ASGPR) expression by selected hepatoma cell lines used in antiviral drug development. Current drug delivery, 5(4), 299-302.

Zhang, L., Tian, Y., Wen, Z., Zhang, F., Qi, Y., Huang, W. & Wang, Y. (2016). Asialoglycoprotein receptor facilitates infection of PLC/PRF/5 cells by HEV through interaction with ORF2. Journal of medical virology88(12), 2186-2195.

Reports that talk about the absence of expression of ASGPR in HeLa cells:

Spiess, M., & Lodish, H. F. (1985). Sequence of a second human asialoglycoprotein receptor: conservation of two receptor genes during evolution. Proceedings of the National Academy of Sciences, 82(19), 6465-6469.

Ji, D. K., Zhang, Y., Zang, Y., Liu, W., Zhang, X., Li, J., ... & He, X. P. (2015). Receptor-targeting fluorescence imaging and theranostics using a graphene oxide based supramolecular glycocomposite. Journal of Materials Chemistry B, 3(47), 9182-9185.

Hu, X. L., Zang, Y., Li, J., Chen, G. R., James, T. D., He, X. P., & Tian, H. (2016). Targeted multimodal theranostics via biorecognition controlled aggregation of metallic nanoparticle composites. Chemical science, 7(7), 4004-4008.

3. The statement "Both BSA-Lac NPs 24 were selectively recognized by ASGPRs as demonstrated by competition assays using an in vitro model of HCC." (lines 24-26, Abstract) is key result of the present study. The experimental results supporting the statement should be described in detail.

In the abstract only the competition assay is mentioned, however, to demonstrate the statement that both BSA-Lac NPs were selectively recognized by the ASGPR, we performed an interaction assay with lactosylated NPs and non-lactosylated NPs. Additionally, the specificity was further confirmed by competition and inhibition assay, which are described in the results section.

The recommendation was taken and in the abstract section it was corrected as follows:

Both BSA-Lac NPs were selectively recognized by ASGPRs as demonstrated by biorecognition, competition e inhibition assays using an in vitro model of HCC.

Also, changes were made in the results section and to improve the explanation, the fluorescence intensity was analyzed quantitatively by confocal microscopy and the representative plot was added.

In the section of the evaluation of specific cellular recognition (results), it was modified as follows:

Evaluation of specific cellular recognition

The cellular interaction of FITC-labeled nanoparticles was studied in two cell lines, HepG2 that abundantly present ASGPR [28,29] and HeLa, that was used as a control, due to the absence of ASGPR [30,31]. The assays were performed by fluorescence confocal microscopy (Figure 5 A) and the fluorescence intensity of cells exposed to NPs were quantitatively evaluated (Figure 5 B).  HepG2 cells incubated with either LC or HC BSA-Lac NPs showed higher fluorescence intensity, indicating that the galactose present in these NPs was recognized by the ASGPR. HepG2 cells incubated with tBSA NPs showed a fluorescence intensity around to 3%, as compared with lactosylated NPs (Figure 5 B). Although it has been reported that albumin is recognized by gp60, a glycoprotein expressed in the membrane of endothelial cells of tumor tissues [6,32], the reduced interaction of HepG2 cells receptors and tBSA NPs can be considered nonspecific.

To confirm the specific recognition of the BSA-Lac NPs with the ASGPR, competition and inhibition assays were done. In the competition assays, a moderate lactosylated NPs uptake was observed (Figure 5 A). The reduced fluorescence intensity (50%, Figure 5 B) is because both the BSA-Lac NPs and free lactose are recognized by the HepG2 ASGPRs. Free lactose present in the media culture competed with the BSA-Lac NPs for the available receptors; thereby reducing BSA-Lac NP receptor binding and resulting in reduced fluorescence. In the inhibition assays, pre-incubation of the HepG2 cells with free lactose show decreased uptake of BSA-Lac NPs (Figure 5 A), reducing approximately 98% of the fluorescence intensity (Figure 5 B). This indicate that previous exposure of the cells to lactose, blocked BSA-Lac NP recognition and support the notion that ASGPRs specifically recognizes the galactose present in BS-Lac NPs [8]. In contrast, as expected, no fluorescence was observed for HeLa cells, because the ASGPR is not expressed by this cell line.

4. The interpretation "Taken together, our studies indicated 99 that BSA-Lac was suitable for the synthesis of nanoparticles with the possibility of selective targeting 100 of hepatic ASGPR." (lines 99-101, the Results section) cannot be driven from the results of FT-IR analysis.

This section was corrected, clarifying that the interpretation was due to all the results obtained in the characterization techniques of lactosylated albumin.

 The results section reads as follows:

 Taken together, results from biorecognition assays of BSA-Lac with RCA, electrophoresis SDS-PAGE and FT-IR, indicated that BSA-Lac was suitable for the synthesis of nanoparticles with the possibility of selective targeting of hepatic ASGPR.

< Minor Points >

1. Statistical analyses for the experimental results is necessary. In addition, the statistical methodology needs to be described in the Methods section.

Statistical analysis was added in the methods section:

Results are presented as mean ± standard deviation. For ELLA, BSA and BSA-Lac assay the Student’s t-test was used. Statistical analysis of analysis of fluorescence intensity of the cellular interaction was performed using a one-way ANOVA followed by the application of Tukey’s test.  P-values < 0.05 were considered statistically significant.

Also, in the figure statistical analysis was added:

Figure 1. Enzyme-linked lectin recognition assays of BSA-Lac and tBSA. Optical density values were compared using the Student’s t-test, p ≤ 0.05. Different letter (a and b) shows statistical significance.

Figure 6. Quantitative analysis of fluorescence intensity of the cellular interaction of FITC-labeled nanoparticles in HepG2 and HeLa cells. Experiments were performed by triplicate and analysis by ANOVA followed by Tukey’s test; p ≤ 0.05. Different letter (a, b, and c) shows statistical significance.

2. The abbreviation should be spelled out, in its first use in the text with the abbreviated form in parentheses.

The abbreviations were spelled out, in its first use in the text.

3. Major and minor grammatical errors are found throughout the text. English editing is absolutely required.

An intensive revision of English was performed.

Reviewer 2 Report

Authors describe the production of lactosylated BSA and its possible use in therapy as hepatocellular-specific carrier The study provide sufficient information on the effective modification of BSA and its main binding to hepatic cells due to the presence of galactose end groups. However, some point need to be clarified:

-the chemical characterization of lactosyl-BSA is incomplete. There is no data about the amount of lactose bound to the protein (in terms of weight percent or number of lactosyl groups). This information should be provided. 

-there is no mention about the procedure followed to label BSA nanoparticles with FITC and how this modification has been quantified. This is important when the binding of different nanoparticles (made of LC or HC BSA-Lac as well as unglycosilated protein)  have to be compared.

-Figure 5 provide only a qualitative observation of the binding of BSA nanoparticles to HepG2 cells. A quantitative analysis of biorecognition, inhibition and competition,  i.e. using a cytometer or at least a densitometric analysis of fluorescence microscopy images, would provide much more information and a better comparison.   

-competition and inhibition analyses reported in figure 5 are negatively affected by the weakness above mentioned. How has been established the amount of free lactose used for the competition and inhibition assays (10 μg in 100 or 200μL). A measure of nanoparticle binding in the presence of increasing concentration of free lactose will provide a better information and the obtained results should be consistent with the amount of lactose bound to the protein.

-legends to figure 4 and 5 are incomplete

some sentences could be modified: line 106 " ..using BSA-Lac and the alcohol..",  line 211 ".. ATR-FTIR spectra? ...", line 216 " ..drop by drop .."  

Author Response

Please find the attached answers to the comments.

Reviewer 3 Report

Figure 1: Optical density is unitless.  The y-axis needs to be revised.  The caption contains "y", which presumably should by "and".

Section 2.1.3 and Figure 3:  The differences in the traces are small, and no reporting of error or statistical significance is provided.  The signal to noise ratio for native BSA is much poorer, suggesting that a lower sample concentration was present.  Such a difference could obscure differences.

Line 111: "Size particle" should be "Particle size"

Line 116: "than" should be "that".

Line 216: "was slowly by drop added" should be "was added slowly by drop".

Author Response

Response to Reviewer 3 Comments

Figure 1: Optical density is unitless.  The y-axis needs to be revised.  The caption contains "y", which presumably should by "and".

The requested changes were made to figure 1.

Section 2.1.3 and Figure 3:  The differences in the traces are small, and no reporting of error or statistical significance is provided.  The signal to noise ratio for native BSA is much poorer, suggesting that a lower sample concentration was present.  Such a difference could obscure differences.

For the ATR-FTIR analysis, an Agilent Cary 630 FTIR Spectrometer was used at a resolution of 4 cm−1 in the range of 650–4000 cm−1.  The samples were lyophilized and the respective powder was analyzed in the spectrometer. The FTIR spectra of the respective samples were highly consistent and the most evident changes were observed in range 900-1199 cm-1 of wavelength corresponding to carbohydrate fingerprint.

Line 111: "Size particle" should be "Particle size"

Change done

Line 116: "than" should be "that".

Change done

Line 216: "was slowly by drop added" should be "was added slowly by drop".

Change done

Round  2

Reviewer 1 Report

The manuscript has been revised well according to my comments.

Reviewer 2 Report

Authors made changes to the manuscript in response to many of the questions asked. Although measurements relating to competition and inhibition have been conducted at a single lactose concentration the manuscript has been improved and can be considered for publication